# A Review of Racial Disparities in Infant Mortality in the US

**DOI:** 10.3390/children9020257

**Published:** 2022-02-14

**Authors:** Caleb J. Jang, Henry C. Lee

**Affiliations:** 1College of Applied Health Sciences, University of Illinois at Urbana-Champaign, Urbana-Champaign, IL 61801, USA; 2Department of Pediatrics, Stanford University, Stanford, CA 94305, USA

**Keywords:** infant mortality, newborn care, healthcare, racism, racial disparities, health disparities, social determinants of health

## Abstract

Racial disparities in infant mortality have persisted, despite the overall decline in the United States’ overall infant mortality rate (IMR). The overall IMR of the entire United States (5.58 per 1000 live births) population masks significant disparities by race and ethnicity: the non-Hispanic Black population experienced an IMR of 10.8 followed by people from Native Hawaiian or Other Pacific Islander populations at 9.4 and American Indians at 8.2. The non-Hispanic White and Asian populations in the United States have the lowest IMR at 4.6 and 3.6, respectively, as of 2018. A variety of factors that characterize minority populations, including experiences of racial discrimination, low income and education levels, poor residential environments, lack of medical insurance, and treatment at low-quality hospitals, demonstrate strong correlations with high infant mortality rates. Identifying, acknowledging, and addressing these disparities must be performed before engaging in strategies to mitigate them. Social determinants of health play a major role in health disparities, including in infant mortality. The study and implementation of programs to address neighborhood factors, education, healthcare access and quality, economic stability, and other personal and societal contexts will help us work towards a common goal of achieving health equity, regardless of racial/ethnic background.

## 1. Introduction

One of the most significant discoveries of the Human Genome Project [1] was that the human nucleotide sequence is nearly identical (99.9%) between any individual regardless of race. Race has been used throughout history to oppress or discriminate against groups, despite stemming from social constructs rather than biological roots. Nevertheless, it is now a categorization in which detection of both long-standing and emerging healthcare disparities can occur. Newborn care and outcomes are no exception. This is evident in a key measure of a society’s health—the infant mortality rate (IMR). Although the IMR in the United States has decreased over the past decade, racial disparities have persisted. It is a moral and healthcare imperative for society to address and discover solutions to eliminate disparities within newborn care.

## 2. Infant Mortality in the United States

Infant mortality has traditionally been defined as the death of an infant before his or her first birthday [2]. More than a statistic, it reflects a tragic moment in a family’s life, and in some instances, the failings of the social and healthcare institutions of a community. The infant mortality rate (IMR) of a population can be determined by measuring the number of infant deaths for every 1000 live births [2]. As of 2020, the United States continues to have one of the highest infant mortality rates (5.58) [3] and was notably higher than other OECD countries. The leading causes of infant mortality include birth defects, preterm birth, low birthweight, complications with maternal pregnancy, sudden infant death syndrome, and injuries [2]. 

The overall IMR of the entire United States population masks significant disparities by race and ethnicity: the non-Hispanic Black population experienced an IMR of 10.8 followed by people from Native Hawaiian or Other Pacific Islander populations at 9.4 and American Indians at 8.2 as of 2018. The non-Hispanic White and Asian populations in the United States (as of 2018) have the lowest IMR at 4.6 and 3.6, respectively [2]. 

A significant decline in IMR among most racial and ethnic populations has been observed in the past several decades. Overall, the IMR in the United States has decreased from 9.2 in 1990 [4] to 5.6 in 2020 [3]. It is promising that minority groups including Black, American Indian, and Hispanic populations have experienced substantial declines in IMR over the past 30 years. However, while the overall IMR has declined, the Black/White disparity in IMR has remained [5]. These disparities are evident when stratifying IMR by race in states throughout the United States. Southern states have a larger proportion of Black individuals in the population than other regions in the United States [6]. These states experienced an average excess of 1.18 infant deaths per 1000 births [7]. This disparity remains even in states with a relatively lower Black population. The state of Colorado, which experienced a relatively low IMR of 4.8 [8], had a Black population that made up only 5.1% of the total population [9]. In 2016, the IMR among Black infants in Colorado was 10.7 (highest among any race) while the IMR for non-Hispanic White infants was 4.0 [10].

Disparities in IMR also exist in the American Indian population. Between 2005 and 2014, all other racial/ethnic populations experienced a decline in IMR except for the American Indian population [11]. Furthermore, IMR among American Indians was highest in states with a greater percentage of American Indians in their overall population. For example, Alaska had the highest American Indian population at 22% as of 2020 [12]. The average IMR from 2016 to 2018 for American Indian infants in Alaska was the highest among any other race/ethnic group at 9.3% [13]. This percentage was more than two times higher than the IMR experienced by White infants in Alaska. 

The Asian/Pacific American (APA) term is used to include those originating from Asian countries as well as Pacific Islands. Consequently, disparities among Pacific Islander groups are often disguised by their categorization as APAs. The average IMR for APAs from 2016 to 2018 was noted to be the lowest among any racial/ethnic group at 4.1 deaths per 1000 live births [13]. However, when separating these two ethnic groups into Asians and Native Hawaiian/Pacific Islanders, the IMR for Asians was 3.6, while the IMR for Native Hawaiian/Pacific Islanders was 9.4 [2]. Pacific Islanders had the second highest IMR with 1.4 fewer deaths than Black families. While there is limited research on this population, current evidence suggests that Pacific Islander women experience a greater risk of hypertension, preterm birth, primary cesarean delivery, preeclampsia, gestational diabetes, and low birth weight [14,15,16]. Further disparities also exist for Pacific Islander subgroups. In a study that tracked infants born in any Hawaii hospital from January 2010 to December 2011, Native Hawaiians had the greatest risk of low birth weight infants, Samoans were at the greatest risk of delivering macrosomic infants and Micronesians were at the highest risk of cesarean delivery [17].

Although Asians have the lowest IMR among the major racial and ethnic categorizations, it is important to note emerging disparities within Asian American subgroups. The Asian Indian subgroup encounters a greater risk for low birth weight and small for gestational age (SGA) births. Studies have shown that infants of the Asian Indian subgroup are five times more likely to be SGA and admitted to the special care nursery than non-Hispanic White infants [18]. Hypothermia, hypoglycemia, and fetal distress are among the adverse conditions that are associated with SGA births—all of which increase the risk of infant mortality [18,19]. A longitudinal study that followed 2.3 million births in California over a 10-year period suggested that the Thai Asian American subgroup experienced the most significant disparities in perinatal outcomes as infants of Thai mothers experienced an IMR 90% higher than non-Hispanic White mothers on a risk-adjusted basis [20].

### Preterm Births

According to the CDC, preterm births (when a baby is born before 37 weeks of pregnancy have been completed) are the second leading cause of IMR. The overall preterm birth rate in the United States has declined from 10.2% in 2019 to 10.1% in 2020 [21]. However, when stratifying preterm birth rates by race, Black and Hispanic populations experience a major disparity. As of 2019, Black families had the highest number of preterm births at 14.4% followed by Hispanic families at 10% [22]. Potential contributing factors to preterm birth include diabetes, heart and kidney disease, hypertension, smoking, inadequate nutrition, incompetent cervix, and premature birth in a previous pregnancy [23]. While these are all behavioral and biological complications that may lead to preterm births, causes due to socioeconomic factors and racism should also be considered when determining strategies to prevent preterm births. 

## 3. Social Determinants of Health and Impact on Infant Mortality

The presence of disparities in infant mortality implicates social determinants of health as a major contributor. Social determinants may be categorized into economic stability, education, healthcare access and quality, neighborhood and built environment, and social and community context [24]. We examine some of the known features of these determinants in their contributions to racial disparities in infant mortality.

## 4. Social and Community Context

Race has been used throughout history to determine privileged versus oppressed groups, which has created invisible barriers for minority groups to overcome, especially in the healthcare system.

Dr. Camara Phyllis Jones, former president of The American Public Health Association, has described how stress caused by external factors such as racial abuse is correlated with adverse health outcomes. She describes three levels of racism: internalized (acceptance by members of the stigmatized races of negative messages about their abilities and intrinsic worth), personally mediated (differential assumptions/actions about the abilities, motives, and intents of others by race) and institutional (differential access to the goods, services, and opportunities of society by race), all of which negatively affect health outcomes [25]. We can consider how these racism-related stressors may have major implications on an individual’s health outcomes even before birth by raising a mother’s stress levels. Elevated stress increases cortisol levels leading to stunted growth and disruption of the immune, vascular, metabolic, and endocrine systems [26]. Among racial/ethnic groups, Black individuals are most prone to experience elevated stress because of racial discrimination [27]. It has been reported that self-reported experiences of racism are associated with increased adverse birth outcomes including low birth weight (LBW), very low birth weight (VLBW), and preterm delivery by three-fold among Black families [28]. A long-term study (2004–2012) by Bower et al. suggested that non-Hispanic Black individuals may be more inclined to experience preterm birth as a result of emotional experiences of racism. Feeling upset by experiences of racism in the 12 months before delivery was significantly associated with greater odds of preterm birth [29].

The high IMR experienced among the American Indian population can also be attributed to historic and persistent degrees of systemic racism. American Indians have been victims of racism predominantly in the scope of policies in the United States. Historically, policymakers in the United States have disregarded American Indians as a separate race by encouraging them to withdraw from their tribe and assimilate with White society. Like Black individuals, they have also been denied basic rights such as the right to vote and attend schools with or marry White people [30]. Consequently, it can be argued that their lack of access to basic human rights has jeopardized this population into becoming victims of various forms of racial discrimination and disparities in adverse health outcomes including their IMR. Findings from a study that surveyed a sample of 3453 American Indian adults ages 18 or older, a quarter of participants reported that they have experienced discrimination when visiting a doctor [31]. Among participants of the study, 15% said that they avoid seeking medical care for themselves or family members due to a fear of being racially profiled. Along with other factors, racism is likely to account for the fact that American Indian mothers who received late or no prenatal care was 2.9 times higher than Whites mothers in 2016 [32]. Current knowledge on Native American health outcomes suggests parallels to Black patient health outcomes with racism as a common denominator [33,34,35]. 

A plethora of evidence demonstrates that a strong correlation exists between race and IMR. Although correlation does not prove causation, the consistent pattern of disparities seen in IMRs for specific populations underscores the importance of considering the effects of racism when examining underlying causes and formulating solutions to mitigate these disparities.

## 5. Healthcare Access and Quality

Maternal care during pregnancy is a vital aspect of optimal fetal development which can influence health outcomes by pathways such as preterm birth, maternal morbidity, and fetal development. A combination of social, economic, environmental, and governmental factors hinders mothers from receiving ideal maternal and fetal care before and during pregnancy and increases the risk of negative fetal and neonatal health outcomes, including the possibility of infant mortality.

### 5.1. Healthcare Insurance

Expansion of affordable healthcare insurance and medical care is another governmental policy that can reduce IMR. Compared to the White population, the American Indian population is more than twice as likely to lack medical insurance. Furthermore, the Indian Health Services (IHS)—a federal agency responsible for providing health services to American Indians and Alaska Natives—is chronically underfunded and facilities often lack services such as emergency departments [36]. Pediatricians make up only 8% of IHS providers and patients experience longer wait times than non-IHS programs for their appointments. 

Between the years 2014 and 2016, a group of states (Medicaid Expansion States) expanded their Medicaid policies allowing for greater coverage of childbirth [37]. In particular, the Affordable Healthcare Act has helped alleviate disparities in health insurance coverage and access to health services by increasing healthcare coverage for Hispanic families and Black families [38]. A study that compared the IMR for Medicaid expansion states and non-expansions states from 2014 to 2016 found that the IMR in non-Medicaid expansion states rose from 6.4 to 6.5 and declined from 5.9 to 5.6 in Medicaid expansion states [39]. As of 2019, nearly 20% of the Hispanic population and 11% of the Black population in the United States were uninsured [40]. More than 90% of those who do not have healthcare coverage are a result of residing in non-Medicaid expansion states, which mostly consist of southern states that have a high Black population [37]. Medicaid expansion adoption in states with high Black and Hispanic populations may have great potential for improving health outcomes, including reducing infant mortality disparities. Table 1 summarizes some of the social determinants of health and their impact on racial disparities in newborn care and health outcomes.

### 5.2. FMLA

Governmental policies play a significant role in reducing IMR. Currently, the United States operates under the Family and Medical Leave Act (FMLA) of 1993, which grants mothers of newborn and newly adopted children 12 weeks of unpaid leave at a company consisting of 50 or more employees. Among OECD nations, the United States was the only country that did not completely provide paid absence for newborn care [45]. While the implementation of FMLA has been associated with a significant decrease in IMR for children of college-educated and married mothers, it has not benefited less privileged individuals, such as those without formal education and single mothers [45,46]. While some states such as San Francisco and New York have established paid leave policies, disparities in the usage of maternity leave programs exist among mothers in the United States. A study that examined the associations between paid maternity leave and maternal/infant health found disparities in the usage of maternity leave among race/ethnicity and income [45]. Black women and low to middle-income women were less likely to utilize maternity leave programs and even if leave was used, the duration of the usage decreased as income decreased. The study also found that women who used maternity leave experienced a 50% decline in the odds of having their infants hospitalized, having themselves hospitalized, as well as seeing a mental health provider. Although this shows the impact of such policies, it is notable that high-income and White households benefited the most from this program as they had greater access and usage rates than non-White populations [47]. To prompt low-income households to utilize this program, the Act was revised to allow for employees to receive up to 8 weeks of paid absence and increased the wage-replacement benefit up to 90% of their gross income [48]. Furthermore, under the Federal Employee Paid Leave Act, federal employees can now (as of 1 October 2020) receive up to 12 weeks of paid absence for birth or adoption/foster care [36]. While these acts have been implemented, they must be carefully monitored for equitable usage. In addition, more states or the federal government should consider adopting a paid leave absence policy that can be equitably implemented to include the most vulnerable populations as this may decrease infant mortality rates and reduce related disparities.

### 5.3. Quality Improvement

Quality of care received at NICUs is essential towards reducing IMR. Black and Hispanic infant deliveries tend to occur in hospitals that are characterized by overall higher rates of morbidity and mortality [41]. When analyzing differences in morbidity and mortality rates in Black, Hispanic, and White very preterm infants among hospitals in New York City, researchers found that Black and Hispanic very preterm infants were born in the highest morbidity and mortality tertile of hospitals [41]. The risk difference for Black compared with White very preterm infants was 20%, and 11% for Hispanic very preterm infants, compared with White very preterm infants [41]. Similarly, in a study that investigated the segregation and quality of care in NICUs by race, results showed that among any racial and ethnic groups, Black families received neonatal care at the lowest quality NICUs [49]. Assurance of high-quality care in NICUs may help reduce racial/ethnic disparities in infant mortality and could be achieved through improvements in hospital policies, cultures, and quality of care practices and through prioritizing resource allocation to hospitals that serve higher-risk populations.

### 5.4. California Perinatal Quality Care Collaborative (CPQCC)

Quality Improvement (QI) networks allow for the detection of disparities in IMR rate among race/ethnic populations and hopefully lead to avenues for action. QI may be particularly suited to address the implementation of standardized practices due to the multidisciplinary nature and complexity of care in the delivery room and NICU. The California Perinatal Quality Care Collaborative (CPQCC), which represents 90% of California’s NICUs, has used QI to track, detect and improve disparities that exist among infants born in California [49]. This program has successfully reduced morbidities using QI such as those for reducing nosocomial infections and rates of necrotizing enterocolitis in California [50]. The CPQCC has also implemented a health equity dashboard that stratifies QI measures and outcomes by race. This dashboard has helped identify a significant disparity in survival without major morbidity and human milk nutrition between Black and White infants, and Hispanic and White infants, respectively [51,52].

### 5.5. Cradle Cincinnati 

Cradle Cincinnati is a network of partners working across sectors to measurably improve preconception health, pregnancy health, and infant health to reduce preterm birth and IMR in Hamilton County, Ohio. The program uses QI science, data sharing, and an “all teach, all learn” to activate change within prenatal care. Through this collaboration and QI, Cradle Cincinnati has been able to identify, address, and improve infant outcomes for babies born in Cincinnati. In one of their baseline reports [53], Hamilton County’s White and Hispanic IMR were aligned with United States’ averages. However, local Black IMR was reported to be more than three times higher than the White rate and 32% worse than the United States’ Black average. QI has not only helped address the disparities that exist in Cincinnati’s NICUs but has also allowed for the construction and implementation of solutions using community health workers, group prenatal care, implicit bias training, and policy-level solutions to reduce the disparity in Cincinnati’s IMR. In their most recent report, they found Black IMR to have been reduced by 33% from an average of 15.8 in (2015 to 2019) to 10.6 in 2020, a 29% reduction in women who lacked prenatal care (2020), as well as a significant reduction in the number of extremely preterm births [54]. They also reported the development and implementation of a campaign that would focus on safe infant sleeping methods, as a follow-through of an observed increase in sleep-related deaths in 2020. The potential to detect, reduce, and improve IMR in the NICU using QI is limitless, and healthcare systems should aspire to utilize QI for this purpose.

### 5.6. Antenatal Corticosteroids (before Childbirth) and Hypothermia

Antenatal corticosteroids (ACS) may be administered to women with a high risk of preterm births. These steroids accelerate the development of the fetal lungs, which can reduce respiratory diseases and further complications that are associated with preterm births. As previously mentioned, Black women are at a greater risk to deliver preterm births than mothers of any other race. However, they receive inadequate care, such as through lower ACS administration rates to mitigate the consequences of preterm birth. For example, a retrospective cohort study that utilized the United States Natality Live Birth database from the CDC (2016–2017) found lower rates of ACS administration among Black patients (37.7%) compared to White patients (44.4%) [42]. Furthermore, in another study that examined whether maternal race is associated with a lower rate of ACS administration in Washington for women at risk of preterm labor, Black women were less likely to receive antenatal steroids, respiratory support, and surfactant to their infants [42]. Hospital quality of care may be a factor that leads to the disparity. The Ohio Perinatal Quality Collaborative found that the rates of antenatal corticosteroid administration remained high in hospitals that systematically monitor ACS usage [55]. Hospitals that participated in the California Perinatal Quality Collaborative project on the topic of ACS for preterm birth were more likely to administer ACS than hospitals that did not participate in the quality improvement collaborative [56]. To reduce the disparate burden of preterm births, appropriate and equitable evidence-based practices should be implemented for this vulnerable patient population.

Hypothermia is a condition associated with intraventricular hemorrhage (IVH) in very low birth-weight infants and can cause harmful infant outcomes including death. One study found that Black infants were more likely to have higher odds of moderate or severe hypothermia compared to any other racial group [57]. While other factors may come into play when analyzing this disparity, a prominent association between care at a low-quality hospital and high rates of hypothermia was found [58].

### 5.7. Maternal Morbidity

Racial disparities in maternal morbidity during childbirth are also present in the United States. In addition to having the highest IMR, maternal morbidity during childbirth also occurs at the highest rate among Black women [59]. Severe maternal morbidity (SMM) can induce short-term or long-term consequences to a woman’s health. While cases of SMM have increased by over 200% over the past 2 years, Black women are most profoundly affected by it and experience SMM 2.1 times more than White women [59]. Black women also experienced greater risks of developing eclampsia, preeclampsia, and postpartum cardiomyopathy [60]. Rates for these three complications were 5 times higher for Black women than for White women [60]. Furthermore, Black women develop obstetric embolism and obstetric hemorrhage 2.3 to 2.6 times higher than White women [60]. These four conditions were all leading causes of maternal mortality in Black women and accounted for 59% of the Black–White maternal mortality disparity [60]. Maternal morbidity can also induce adverse infant health outcomes. One study found that severe maternal morbidity was associated with a greater risk of delivering very preterm live births and that Black mothers were at the greatest risk to severe maternal morbidity complications [61].

### 5.8. The Power of Language 

Disparities also prevail in the representation of racial/ethnic backgrounds of employees within the healthcare workforce. Healthcare does not have adequate representation of diverse racial and ethnic backgrounds of patients in the United States. According to a report by the AAMC, among all active physicians in the United States, half identified as White and only 5% identified as Black [62]. Being in an environment that a patient is familiar with regarding their language and culture can have a significant effect on health outcomes. A study assessing the effect of language on health discovered that Spanish-speaking parents were four times more likely to incorrectly identify their child’s diagnosis, had fewer physician encounters, and were given fewer updates compared to English-speaking parents [63]. Overall, there is a lower representation of Black and Hispanic healthcare providers in NICUs across the United States [64]. With the prediction of a shortage in the healthcare workforce within the next decade [65], partnering with and creating student pipeline programs that expose underrepresented minority populations to healthcare-related professions to increase the level of racial representation in the healthcare workforce must be considered.

### 5.9. Family Engagement

Another disparity that is prevalent in the NICU derives from what may be considered a non-clinical form of care—family engagement. Kangaroo care (skin-to-skin holding), visitation, traditional holding, and infant massaging are family engagement techniques that are associated with positive health outcomes [66]. Studies have shown that Kangaroo care is associated with stabilized oxygen saturation levels and heart rates (analgesic effect), increased regular breathing patterns, breastfeeding success, and positive neurodevelopmental outcomes [66]. Infants of non-White, socioeconomically disadvantaged, and publicly insured mothers experience a lower rate of family engagement in the NICU compared to their White counterparts [67]. Because hospitals operate under different policies and regulations, it is difficult to pinpoint an exact cause of this disparity. Nevertheless, identifying barriers and creating policies that encourage family visitations to improve infant outcomes is essential.

## 6. Education

Lower parental education levels are correlated with an increased risk of infant mortality. A study that examined education levels and IMR reported that mothers who were not able to attain a high school education (less than 12 years of education) experienced a greater risk of infant mortality by 2.4 times [5]. Hispanic, American Indian, and Black infants are at the greatest risk for mortality, with these populations having the highest percentages of adults with no high school diploma [68]. However, even after accounting for education, disparities remain by race. Although the IMR for the groups of mothers with <12 years of education has decreased substantially by 49% over the past 30 years [5], the IMR for Black infants (10.2 per 1000 live births) was almost twice that of White infants (5.4 per 1000 live births) among populations that achieved a similar level of education [69]. Furthermore, when comparing IMR among highly educated Black families and less educated White families, Black infants still had a higher IMR (46% higher) [5]. This suggests that while greater access to education may improve IMR amongst at-risk populations and is a worthwhile societal goal, it will only partially contribute to reducing the disparity in IMRs among racial groups.

## 7. Economic Stability

Lower-income levels and associated factors are also correlated with an increased risk of preterm birth. Redlining, poor environmental, and suboptimal residential conditions are primary factors that lead to higher risks of preterm births. High-income Black households share similar rates of unemployment, educational attainment, poverty, and single-headed households with low-income White households [43]. This propensity may be due to Black families possessing a greater preference to reside in neighborhoods with higher proportions of Black families, which tend to be located in poor urban/suburban environments. Furthermore, while the legal practice of redlining (the practice of denying a loan for housing in neighborhoods deemed “poor”) has been abolished, the effects still remain and have caused underdeveloped neighborhoods to receive a low Home Owner’s Loan Corporation (HOLC) score. A study [44] that assesses if redlining was associated with risk of preterm discovered that infants born in areas with a low HOLC grade D (hazardous) were both relatively and more likely to be preterm compared to areas with a high HOLC grade A (best). Redlining is a notable contributor to the large burden of infant mortality carried among minority populations.

## 8. Neighborhood and Built Environment

Primary care physicians (PCPs) play an important role in reducing IMR by providing additional prenatal screenings and referring mothers to specialized care (Ob-gyns) [70]. States that have higher ratios of PCPs had lower rates [37] of infant mortality. PCP shortages and closed hospitals are also commonly found in predominantly Black neighborhoods. A study that examined the association between residential segregation and geographic access to primary care physicians in metropolitan statistical areas revealed a correlation between the degree of segregation and the odds of being a PCP shortage area for many Black residential areas [71]. This could partly explain why many Black women lack a PCP, thereby increasing the risk of poor healthcare delivery during pregnancy and potentially leading to a higher risk of adverse infant outcomes, including mortality [72]. Geographic racial segregation is also associated with a disparity at the level of quality care in NICUs [49]. One study described how Black families were more likely to receive care for their very low-birth-weight and very preterm infants at lower-quality NICUs compared to other racial groups [49]. In this study, NICU quality was assessed using Baby-MONITOR (Measure of Neonatal Intensive Care Outcomes Research) scores that relate to nine infant-level measures: antenatal steroid exposure; hypothermia on admission; non-surgically induced pneumothorax; healthcare-associated bacterial or fungal infection; chronic lung disease; timely retinal examination; discharge on human breast milk; mortality during the birth hospitalization; and growth velocity. This study showed that 6% (*n* = 2022) Black infants were cared for in regions with high Baby-MONITOR scores and 76% (*n* = 9686) received care at NICUs with the lowest scores. This disparity is partly due to residential segregation, which manifests from the inherent structural racism rooted in history and persisting today.

Predominantly Black neighborhoods are also characterized as having higher allergen counts as well as pollution levels of almost 1.54 times higher than the overall population [73]. Multiple studies have indicated that pollution can increase the risk of preterm births and lead to the development of unfavorable health conditions such as diabetes [74,75]. The susceptibility of Black mothers to having preterm births is likely contributed to by their tendency to reside in relatively unfavorable environments.

## 9. Conclusions

The prevalence of racial disparities in newborn care is a prominent issue in healthcare that needs to be resolved. To reduce these disparities, we as a society must begin by looking at the discrepancies that lie within social determinants that are associated with adverse infant outcomes. Social determinants may be categorized into economic stability, education, healthcare access and quality, neighborhood and built environment, and social and community context.

Solutions to mitigate these disparities may be achieved through the implementation of policies and practices at multiple levels of society and the healthcare system. Assessing the effectiveness and reproducing solution mechanisms that mirror successful interventions can help reduce the racial/ethnic disparities in IMR as we work towards a common goal of achieving health equity, regardless of racial/ethnic background. 

## Figures and Tables

**Table 1 children-09-00257-t001:** Social determinants of health and impact on racial disparity in newborn care and outcomes.

Social Determinant	Disparities
Social and community context	Black individuals are more inclined to experience preterm birth as a result of emotional experiences of racism [28]. In a survey of 3452 American Indian adults ages 18 or older, a quarter of participants reported that they have experienced discrimination when visiting a doctor [31].
Healthcare access and quality	As of 2019, nearly 20% of the Hispanic population and 11% of the Black population in the United States were uninsured [40]. When analyzing differences in morbidity and mortality rates in Black, Hispanic, and White very preterm infants among hospitals in New York City, researchers found that Black and Hispanic very preterm infants were born in the highest morbidity and mortality tertile of hospitals [41]. Antenatal corticosteroids are administered to Black patients at a lower rate despite them being at a greater risk for delivering preterm births [42].
Education	Hispanic, American Indian, and Black infants are at the greatest risk for mortality, with these populations having the highest percentages for adults who did not have a high school diploma [5]. Mothers who were not able to attain a high school education (less than 12 years of education) experienced a greater risk of infant mortality by 2.4 times [5].
Economic stability	High-income Black households share similar rates of unemployment, educational attainment, poverty, and single-headed households with low-income White households [43]
Neighborhood and built environment	Neighborhoods that are predominantly Black are characterized as having higher allergen counts as well as pollution levels of almost 1.54 times higher than the overall population [44].

## Data Availability

Not applicable.

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
