# Peer review of "A Review of Racial Disparities in Infant Mortality in the US"

_children, 2022, doi:10.3390/children9020257_

Round 1
Reviewer 1 Report
Racial Disparities in Newborn Care among US-born Infants
This manuscript is a reasonably comprehensive review of non-biological factors that may contribute to long-standing racial-ethnic disparities in infant mortality in the US. I have put in 3 types of reactions: overall weaknesses, minor weaknesses, and specific comments.
General weaknesses: The major problem I find is that the authors have not managed an objective view of the problem (racism), and how it plays out, particularly for IM. This weakens the impact of their message. Examples (there are many):
- A more objective version of line 16 might be “Race has been used throughout history to determine privileged vs. oppressed or discriminated against groups…”.
- “Racism has continued to play a significant role in promoting racial and ethnic health disparities among US born infants.” is a strong statement which is not backed up by evidence until much later in the paper.
- “These statistics could account for the fact…” does weaken the assertion with the word “could,” but more importantly there are many intervening issues that would need to be considered to assert causality.
Other weaknesses: Although overall the paper is well written, a thorough review is needed by an experienced editor. Examples:
- Simple grammar: line 251 is missing an article between “that” and “IMR.” “…industries especially the healthcare workforce is deficient…” is missing commas and agreement in verb tense. Although relatively minor, the frequency of these errors distracts the reader and hinders the ability to take your message as seriously as it merits.
- Lines 219 and 222 have examples of proofreading errors. There are many more.
- Imprecise phrases like “…has increased five years in a row, including most recently from…” (line 265). Subjective terms like “Evidently” and “Unfortunately” do not reflect good scientific writing.
- Unnecessary capitalization: “The” is not actually part of the Human Genome Project’s official name. “States” (line 51) and “Southern States’ (line 60) are not names.
- Except for the title, “US” should be spelled out each time.
- Minor point: when talking about “risk,” it is not necessary to use qualifiers–risk does not mean certainty. For example, “can” is not needed in line 88; adverse conditions *do* impart increased *risk*.
- Evidence to back up an assertion should ideally be as close to the assertion as possible, rather than a sentence or two later. For example, “Black women are at a greater risk to deliver preterm 279 births than mothers of any other race, and yet receive inadequate care through lower ACS 280 administration rates to alleviate this risk.” The immediate reaction upon reaching the period is “What evidence?” The “inadequate care” clause would be better made incorporated into the next sentence, with the evidence.
Specific comments:
Lines 29 and 45: CDC Wonder has IMR data through 2020.
Line 84: I am unfamiliar with “Asian American Indian” as a subgroup.
Line 130: Correlation is not causation. Correlations are not *suggested* between racism and IMR–they are clearly documented in reputable data. The question is whether the associations are causal. The divorce rate and the consumption of butter are *correlated* but unlikely to be causal.
Line 169: Many readers may be unfamiliar with the term “redlining.” Susceptible (line 171) is not the best choice of word for this sentence. Please note that only the *legal* practice of redlining has been abolished.
Line 189: This should probably start a new subsection.
Line 225: This does not seem to be an accurate synopsis of a complicated analysis. Here, you have compared the effect of a California policy to states without the policy. Greater detail would be needed to talk about anything other than a pre/post effect *in California.* Also, post-neonatal mortality is not the same as IMR, and have quite different actual causes of death.
Line 347: This section mixes describing the problem with cursory discussions of potential solutions, which possibly would be better in the QI section.
Line 445: Who is "we"? "we the people” or “we the authors”?
Author Response
Ref.: Children-1558081
Racial Disparities in Newborn Care among US-born Infants: Review
Children (MDPI)
Dear Kelly Qiao,
Thank you for your kind consideration of our manuscript entitled “Racial Disparities in Newborn Care among US-born Infants: Review.” We are grateful for the Editor’s and Reviewer’s thoughtful comments. We have addressed the concerns in the manuscript and our response follows:
Reviewer comment 1: The major problem I find is that the authors have not managed an objective view of the problem (racism), and how it plays out, particularly for IM. This weakens the impact of their message. Examples (there are many):
- (1) A more objective version of line 16 might be “Race has been used throughout history to determine privileged vs oppressed or discriminated against groups…”.
- (2) “Racism has continued to play a significant role in promoting racial and ethnic health disparities among US born infants.” is a strong statement which is not backed up by evidence until much later in the paper.
- (3) “These statistics could account for the fact…” does weaken the assertion with the word “could,” but more importantly there are many intervening issues that would need to be considered to assert causality.
Author Response:
(1) We have revised the paper to portray more of an objective perspective of how racism affects infant mortality. Line 16 has been revised to “Race has been used throughout history to determine privileged vs oppressed groups which have created invisible barriers for minority groups to overcome, especially in healthcare.”
(2) We have reorganized the first half of the second paragraph in section 2 (Infant Mortality in the United States) to become the first paragraph of section 3 (Societal Impact of Racism on Perinatal Health). This paragraph touches upon the role racism plays in promoting racial disparities in newborn care which is the main discussion point of section 3.
(3) To acknowledge other factors, along with racism, that can affect poor infant outcomes, we have revised lines 125-127 to: “Along with other factors, racism is likely to play a significant role in the prominent disparity between non-Hispanic White and American Indian mothers (2.9 times higher) who receive late or no prenatal care.”
Reviewer comment 2: Other weaknesses: Although overall the paper is well written, a thorough review is needed by an experienced editor.
Examples:
- (1) Simple grammar: line 251 is missing an article between “that” and “IMR.” “…industries especially the healthcare workforce is deficient…” is missing commas and agreement in verb tense. Although relatively minor, the frequency of these errors distracts the reader and hinders the ability to take your message as seriously as it merits.
- (2) Lines 219 and 222 have examples of proofreading errors. There are many more.
- (3)Imprecise phrases like “…has increased five years in a row, including most recently from…” (line 265). Subjective terms like “Evidently” and “Unfortunately” do not reflect good scientific writing.
- (4) Unnecessary capitalization: “The” is not actually part of the Human Genome Project’s official name. “States” (line 51) and “Southern States’ (line 60) are not names.
- (5) Except for the title, “US” should be spelled out each time.
- (6) Minor point: when talking about “risk,” it is not necessary to use qualifiers–risk does not mean certainty. For example, “can” is not needed in line 88; adverse conditions *do* impart increased *risk*.
- (7) Evidence to back up an assertion should ideally be as close to the assertion as possible, rather than a sentence or two later. For example, “Black women are at a greater risk to deliver preterm 279 births than mothers of any other race, and yet receive inadequate care through lower ACS 280 administration rates to alleviate this risk.” The immediate reaction upon reaching the period is “What evidence?” The “inadequate care” clause would be better made incorporated into the next sentence, with the evidence.
Author Response:
- We appreciate these comments and careful reading of our manuscript. We have improved the writing, including adding “the” between “that” and “IMR” (“that the IMR”) and have also restructured lines 350-352 to say, “The healthcare workforce is deficient in being representative of the diverse racial and ethnic backgrounds of patients in the United States. According to a report by the AAMC, among all active physicians in the United States, half identified as White and only 5% identified as Black.” We have also proofread the document overall and made other similar corrections.
- Line 219 has been modified to “Disparities in the usage of maternity leave programs exist among mothers in the United States.” And line 222 has been modified to “60% of mothers in the United States have access to unpaid leave. However, due to a fear of losing their job as well as financial constraints, many mothers choose not to utilize this benefit.”
- The phrase in line 265 has been removed along with the statistic. This has been replaced with a new statistic that mentions the most recent data on the preterm birth rate in the United States. It now says, “The overall preterm birth rate in the United States has declined from 10.23% in 2019 to 10.09% in 2020. However, when stratifying preterm birth rates by race, Black and Hispanic populations experience a major disparity.” Furthermore, the subjective terms (“evidently” and “unfortunately”) have been removed from the paper.
- The “The” in the The Human Genome Project has been corrected to a lowercase letter (t). “States” (line 51) and “Southern States” have also been replaced with lowercase letters.
- The term “US” has been spelled out to “United States” each time it was mentioned except for the title.
- Qualifiers used when talking about “risk” have been removed. Line 88 has been corrected to “Hypothermia, hypoglycemia, and fetal distress are among the adverse conditions that are associated with SGA births — all of which increase the risk of infant mortality.”
- In terms of addressing evidence to back up the assertion made in line 279 regarding the high risk of inadequate care for preterm births among Black women, we start to describe right after evidence to back up this statement with descriptions of several retrospective cohort studies. For example, we report on a study that revealed that Black patients receive lower ACS administration than White patients even though they are at a greater risk to deliver preterm births. To improve flow as the reviewer has pointed out,, we have revised lines 297-283 to, “As previously mentioned, Black women are at a greater risk to deliver preterm births than mothers of any other race. However, they receive inadequate care, such as through lower ACS administration rates to mitigate the consequences of preterm birth. For example, a retrospective cohort study that utilized the United States Natality Live Birth database from the CDC (2016-2017) found lower rates of ACS administration among Black patients (37.7%) compared to White patients (44.4%).”
Reviewer comment 3: Specific comments:
- (1) Lines 29 and 45: CDC Wonder has IMR data through 2020.
- (2) Line 84: I am unfamiliar with “Asian American Indian” as a subgroup.
- (3) Line 130: Correlation is not causation. Correlations are not *suggested* between racism and IMR–they are clearly documented in reputable data. The question is whether the associations are causal. The divorce rate and the consumption of butter are *correlated* but unlikely to be causal.
- (4) Line 169: Many readers may be unfamiliar with the term “redlining.” Susceptible (line 171) is not the best choice of word for this sentence. Please note that only the *legal* practice of redlining has been abolished.
- (5) Line 189: This should probably start a new subsection.
- (6) Line 225: This does not seem to be an accurate synopsis of a complicated analysis. Here, you have compared the effect of a California policy to states without the policy. Greater detail would be needed to talk about anything other than a pre/post effect *in California.* Also, post-neonatal mortality is not the same as IMR, and have quite different actual causes of death.
- (7) Line 347: This section mixes describing the problem with cursory discussions of potential solutions, which possibly would be better in the QI section.
- (8) Line 445: Who is "we"? "we the people” or “we the authors”?
Author Response:
- To use the most recent data regarding IMR in the United States, we have now removed the 2019 statistic and replaced it with the 2020 statistic (5.58 births/1,000) deaths. This correction has been made in lines 29 and 45.
- The term “Asian American Indian” used in lines 84 and 85 have been corrected to “Asian Indian.”
- We agree that correlation does not prove causation. However one of the objectives of this paper is to address the impact race/ethnicity has on infant mortality. As a result, we have revised line 130 to say, “A plethora of evidence demonstrates a strong correlation that exists between race and IMR. Although correlation does not prove causation, the consistent pattern of disparities seen in IMRs for specific populations must consider the effects of racism when examining underlying causes and formulating solutions to mitigate these disparities.”
- To consider the unfamiliarity of the term “redlining” (line 169) we have defined it after it is mentioned in parentheses (the practice of denying a loan for housing in neighborhoods deemed “poor”). Furthermore, the word “susceptible” has been removed in line 171 to now say, “High-income Black households share similar rates of unemployment, educational attainment, poverty and single-headed households with low-income White households.”
- We have included a new sub-section titled “Primary Care Physician Usage and Access” to fit under section 4 (Socioeconomic Factors and Infant Mortality). We agree that this helps the organizational aspect of the paper for better communication to our readers.
- Upon reviewing the article that discussed the effect California’s paid leave policy had on birth outcomes in California, the authors of the article discovered that in addition to post-neonatal mortality, infant mortality also dropped upon the implementation of California’s paid leave policy. As a result, we have now modified lines 224-255 to say, “Upon the implementation of the statute, California’s infant mortality rate dropped.”
- Line 347 has been moved to section 7 (Ethnic/Racial Representation in the Healthcare Workforce) due to the content describing a problem and discussing possible solutions which are also discussed in section 7.
- The pronoun “we” used in line 445, has been clarified and revised to “we as a society.”

Reviewer 2 Report
Thank you for the opportunity to review the manuscript titled, “Racial Disparities in Newborn Care among US-born Infants”. The authors have generated an evidence base to understand the racial and socioeconomic disparities in newborn care in US. This is a critically important research topic. This research includes many different constructs that are well described and presented. Please find the following minor comments for the revision.
Abstract: Abstract should reflect the key evidence synthesised from the review article. Please summarise the findings of key concepts of the review.
Introduction: Rationale for the review should be elaborately justified.
Authors are advised to add a table summing up descriptive statistics in racial and socioeconomic disparities in newborn care, as it may be eye catchy for the readers and may increase readability of the article.
“Section 3. Societal Impact of Racism on Perinatal Health” is impressively written.
VLBW on itself may be a section in the article.
Conclusion: Well written and summarised.
Author Response
Ref.: Children-1558081
Racial Disparities in Newborn Care among US-born Infants: Review
Children (MDPI)
Dear Kelly Qiao,
Thank you for your kind consideration of our manuscript entitled “Racial Disparities in Newborn Care among US-born Infants: Review.” We are grateful for the Editor’s and Reviewer’s thoughtful comments. We have addressed the concerns in the manuscript and our response follows:
Reviewer comment 1: Abstract: Abstract should reflect the key evidence synthesised from the review article. Please summarise the findings of key concepts of the review.
Author Response: To reflect key evidence synthesized from our article we have added to the abstract: “The overall IMR (per 1,000 births) of the entire United States (5.58) population masks significant disparities by race and ethnicity: the non-Hispanic Black population experienced an IMR of 10.8, followed by people from Native Hawaiian or Other Pacific Islander populations at 9.4, and American Indians at 8.2. The non-Hispanic White and Asian populations in the United States have the lowest IMR at 4.6 and 3.6, respectively. A variety of factors that characterize minority populations including, experiences of racial discrimination, low income and education levels, poor residential environmeents, lack of medical insurance, and treatment at low-quality hospitals demonstrate strong correlations with high infant mortality rates. Identifying, acknowledging, and addressing these disparities must be performed in order to engage in strategies to mitigate them. Through the implementation of policies and practices at multiple levels of society and the healthcare system, healthcare disparities can be reduced by working towards a common goal of achieving health equity.”
Reviewer comment 2: Introduction: Rationale for the review should be elaborately justified.
Author Response: To justify our rationale for writing this review, we have added a statement that discusses what previous studies include as well as how our study is unique in that we combine racial disparities in newborn care from a variety of sources as well as provide solutions to mitigate these disparities. A statement at the end of the introduction paragraph has been added: “Previous studies have been performed to address specific racial/ethnic disparities among infants born in the United States; this paper seeks to combine prominent factors that further promote racial/ethnic disparities in newborn care, as well as propose a variety of strategies to work toward reducing these disparities.”
Reviewer Comment 3: Authors are advised to add a table summing up descriptive statistics in racial and socioeconomic disparities in newborn care, as it may be eye catchy for the readers and may increase readability of the article.
To enhance readability of our paper, we have included a table that outlines the key points mentioned from section 4 (Socioeconomic Factors and Infant Mortality). These key points include the major racial disparity mentioned from each socioeconomic factor.
​​Table 1. Socioeconomic factors and impact on racial disparity in newborn care and outcomes.
|
Socioeconomic Factor |
Disparity |
|
Education |
Hispanic, American Indian, and Black infants are at the greatest risk for mortality, with these populations having the highest percentages [28] for adults who did not have a high school diploma.
Singh and Yu’s study [27] that examined education levels and IMR reported that mothers who were not able to attain a high school education (less than 12 years of education) experienced a greater risk of infant mortality by 2.4 times. |
|
Environment |
Neighborhoods that are predominantly Black are characterized to have higher allergen counts as well as pollution levels of almost 1.54 times higher [34] than the overall population. |
|
Primary Care Physician |
As the degree of residential segregation increases, the odds of having a primary care physician shortage area increases for majority Black zip codes. |
|
Governmental Policies |
Compared to the White population, the American Indian population is more than twice as likely to lack medical insurance.
More than 90% [49] of those who do not have healthcare coverage are a result of residing in non-Medicaid expansion states, which mostly consist of southern states that have a high Black population. |
Reviewer comment 4: VLBW on itself may be a section in the article.
To enhance the readability of our paper we have separated the subsection (Maternal Morbidity and Cesarean Delivery During Childbirth) in Section 6 (Hospital Care) to two subsections titled “maternal morbidity” and “Cesarean Delivery and VLBW infants.” VLBW is discussed in tandem with cesarean deliveries, which is why we chose not to dedicate a subsection to “VLBW.”
